# Biodegradable Active Packaging Material Containing Grape Seed Ethanol Extract and Corn Starch/κ-Carrageenan Composite Film

**DOI:** 10.3390/polym14224857

**Published:** 2022-11-11

**Authors:** Cuntang Wang, Xuanzhe An, Yueyi Lu, Ziyu Li, Zengming Gao, Shengxin Tian

**Affiliations:** College of Food and Bioengineering, Qiqihar University, Qiqihar 161006, China

**Keywords:** grape seed ethanol extract, corn starch, κ-carrageenan, antioxidant, lard packaging, active film

## Abstract

An active film composed of corn starch/κ-carrageenan and ethanolic grape seed extract (0, 1, 3, and 5 wt% of GSE on corn starch basis) were successfully prepared using the solvent casting technique. The effects of the different concentrations of ethanolic grape seed extract (GSE) on the physicochemical properties, antioxidant properties, and antibacterial properties of CS/κC films were analyzed. The results showed that the addition of GSE inhibited the recrystallization of starch in the composite film. The glass transition temperature of composite film is 121.65 °C. With the addition of GSE, the surface roughness of the composite film increased, and the cross-section displayed a stratification phenomenon. Meanwhile, when GSE was added to the composite film, the tensile strength of the composite film decreased (3.50 ± 0.27 MPa), the elongation at break increased (36.87 ± 2.08%), and the WVP increased (1.58 ± 0.03 g mm/m^2^·d· kPa). With the increase of the concentration of GSE in the composite film, the a* value and b* value of the composite film increase, the L* value decreases, and the opacity increases. The lipid oxidation test proved that the composite films containing 1% GSE has a significant inhibitory effect on the oxidation of lard (*p* < 0.05). The above results indicate that the GSE can be used as a food-grade packaging material and has a good application prospect in the food industry.

## 1. Introduction

Today, as a product of petrochemical industry, plastic is widely used in food packaging because of its low price and convenient use [1]. However, plastics are not easy to degrade, and the harmful gases produced after incineration can significantly pollute the soil, water, and the atmosphere, ultimately endangering human health. To this end, safe and biodegradable biological films are gradually being developed to replace plastic films to reduce environmental pollution [2,3]. At present, biodegradable materials, such as polysaccharides (starch, cellulose, and chitosan), proteins (soy protein, corn zein, and whey protein), and lipids (paraffin, acetoglyceride, and shellac resins) have been used to prepare biodegradable films [4,5]. Starch has the advantages of low price, non-toxic, comprehensive sources, renewable, and good film-forming properties, and it is therefore used to develop naturally biodegradable bio-based films as one of the significant plastic alternatives [6,7,8]. Moreover, corn silk polysaccharide has a good immunomodulatory effect [9].

Although starch-based materials have good oxygen barrier properties, the use of corn starch to produce biodegradable films has disadvantages, such as poor moisture resistance and poor mechanical properties, and its application is limited [10]. To overcome these shortcomings, mixing starch with other biopolymers to form composites has received much attention, such as carrageenan (CA), chitosan, gelatin, xanthan gum sodium, carboxymethyl cellulose, etc. CA mainly derived from algae and is a water-soluble polymer consisting of a linear chain of partially vulcanized galectins, which has been widely used in food gelation, emulsification, and thickening [11,12]. In addition, CA also has excellent gelatinization and high viscosity characteristics and has good film-forming properties. When blended with starch, the mechanical properties of starch film can be improved. Sandhu et al. [13] blended pearl rice starch and carrageenan to prepare composite film and found that the addition of carrageenan could improve the tensile strength of the film, reduce the water vapor transmission rate, and improve the physical properties of the film.

However, starch-carrageenan composite membrane is lacking in biological activity; therefore, natural active agents can be introduced to improve the antioxidant and antibacterial properties of composite membrane. Polyphenolic components, such as phenolic acids, flavonoids, and anthocyanins, are usually added to starch composite film to improve the antioxidant and antimicrobial properties of food packaging film, so as to prevent food spoilage and prolong the shelf life [14].

Grape pomace, as a solid residue in wine making and grape juice industry, accounts for about 20~25% of the quality of grapes used [15]. Grape pomaces are traditionally composted or used as animal feed, but the polyphenols in the pomace are wasted. The exploitation and utilization of grape pomes can not only optimize the environment and save resources but also create good social economic and social benefits. It has been reported that grape pomace extract is rich in polyphenols, such as catechin, epicatechin, gallic acid, and proanthocyanins, which show a variety of biological activities, such as good antibacterial, antioxidant, and anti-inflammatory activities [16,17,18]. Grape seed ethanol extract is added to the film, which can effectively reduce the growth rate of bacteria and the oxidation rate of fat and keep the product color stable when stored at low temperature [19]. M. Gomaa et al. found that biodegradable films prepared by GA and AOE could prolong the shelf life of Agaropsis bisporus [20]. Abdin et al. added Syzygium cumini seeds extract (SCSE) to sodium alginate/gum arabic films (SG) to prolong the shelf life of canola oil [21].

Therefore, grape seed extract can be added to films as antioxidant material to prepare food-active packaging and to develop a biodegradable antioxidant film. However, there are few reports on preparing composite films by blending grape seed extract with corn starch and CA.

Therefore, the purpose of this study is to prepare environmental-friendly and functional CS/κC/GSE composite films with the different concentrations of GSE as raw materials. In addition, the effects of the different contents of GSE on the properties of CS/κC films are also evaluated to select a good food-packaging film. It is especially important that the characterization of CS/κC/GSE composite films investigates the effects of GSE on film properties, including physicochemical, mechanical, microstructure structure, antioxidant, and antibacterial activities. We hope to develop an environmentally friendly, biodegradable, and bioactive thin film material. This is the first attempt to develop biodegradable films using CA/glycerol/CS and GSE, which could improve the oxidative stability of food and prolong the shelf life of food.

## 2. Materials and Methods

### 2.1. Materials

Corn starch was obtained from Heilongjiang Yufeng Corn Development Co., Ltd. (Qiqihar, China). Grape seeds and pig lard were obtained from Qiqihar Liuyuan market, Carrageenan (food grade, Qingdao Dehui Marine Biotechnology Co., Ltd., Qingdao, China). Yeast Extract and Trypsin (biochemical reagent) were obtained from Beijing Aoboxing Biotech Co., LTD. *S. aureus* strains (ATCC29213) and *Escherichia coli* (ATCC25922) were obtained from School of Food and Biological Engineering, Qiqihar University (Qiqihar, China). 2,2-Diphenyl-1-picrylhydrazyl (DPPH) and 2,2′-azino-bis-(3-ethylbenzothiazoline-6-sulfonic acid) (ABTS) were supplied by Sigma-Aldrich Chemical Co. (St. Louis, MO, USA). Other chemical reagents (analytical grade) were obtained from Tianjin Kaitong Chemical Reagents Co., LTD., (Tianjin, China) We declare that all experimental materials and methods are in compliance with the relevant laws and regulations of the People’s Republic of China.

### 2.2. Extraction of GSE

GSE extraction was performed according to the method of Gao et al. [22] with some modifications. The grape seeds were crushed with a multi-function (GX-220 multi-function crusher, Zhejiang High-tech Industry and Trade Co., LTD.) and 70% ethanol was added to the crushed grape seeds at a ratio of 1:20 (W: V). After extraction at room temperature for 2 h, the extract solution was centrifuged at 6000× *g*, and the extraction residue was repeated the above extraction method twice. Then, the three supernatants were combined, transferred to a rotary evaporator (2L-ARE rotary evaporator, Shanghai Haozhuang Instrument Co., Ltd., Shanghai, China) and concentrated at 50 °C. The concentrate was lyophilized in a vacuum freeze dryer (2.5 L-FreezePrySystem, Labconco, Kansas City, MO, USA) to obtain GSE powder, which was stored in a refrigerator at 4 °C. The extraction yield of GSE in grape seed was 9.24%.

### 2.3. Preparation of Films

CS/κC-GSE films were prepared by solution casting according to the method of Riaz et al. [5] with slight modifications. Corn starch gelatinized solution was prepared by dissolving 9 g CS in 297 g distilled water and gelatinizing it for 30 min at 90 °C. Then, 1.8 g κ-carrageenan was added into CS gelatinized solution with stirring for 30 min at 90 °C. After that, different weights of GSE (0, 1, 3, and 5 wt% on CS basis) were added into CS/κC film-forming solutions with stirring 30 min at 90 °C. Then, 1.8 g glycerol was added as a plasticizer to the CS/κC-GSE film-forming solution and stirred for 30 min at 90 °C, using hot water to complement the evaporated water. The CS/κC-GSE film-forming solution was sonicated at 90 °C for 30 min to remove bubbles. A 120 mm diameter polyethylene ring was fixed on a glass plate covered with release paper. The CS/κC-GSE film-forming solutions (25 g) were poured onto the polyethylene ring and fixed for 15 min. The glass plates with film-forming solutions were placed in a blast drying oven and dried at 40 °C for 12 h. Then, the films were carefully removed from the glass plates. The films were placed in a dryer at 25 °C with a relative humidity of 56.8% (NaBr saturated solution) and were balanced for 72 h to determine the parameters of the films. The films were placed in a desiccator with a relative humidity of 57.57% (NaBr saturated solution) at 25 °C for 72 h and then the indexes of the film were determined. Finally, the prepared composite films containing 0, 1, 3, and 5 wt% GSE were designated CS/κC, 1% CS/κC-GSE, 3% CS/κC-GSE, and 5% CS/κC-GSE films, respectively.

### 2.4. Characterization of the CS/κC-GSE Films

#### 2.4.1. Mechanical Property

The mechanical properties of composite films were determined by referring to the method of Huang et al. [23], with minor modifications. This experiment used a texture analyzer (TA. XT plus C, Stable Micro System Co., Godalming, UK) to measure the mechanical properties of the film. The film samples are trimmed to a 6 cm × 2 cm rectangle. The test speed was 2 mm/s and the initial clip distance was 20 mm. The tensile strength (MPa) and elongation at break (%) of the composite film were measured. It is averaged after three measurements of each sample. Tensile strength (TS) and elongation at break (EB) of the composite film are calculated as follows:(1)$$\mathrm{TS}=\frac{F_{\max}}{S}$$
(2)$$\mathrm{EB}\%=\frac{L_{1}-L_{0}}{L_{0}}\times 100\%$$

In the formula: TS is the tensile strength (MPa), F_max_ is the maximum load at film fracture (N), S is the cross-sectional area of the film (mm^2^), EB is the elongation at break (%), L_1_ is the length of the film after stretching (mm), and L_0_ is the initial length of the film (mm).

#### 2.4.2. Water Vapor Permeability (WVP)

The WVP was determined by weight and referring to the method of Roy et al. [24] with a slight modification. The 10 g of anhydrous CaCl_2_ was placed in a blast air oven and dried at 110 °C for 2 h and placed it in a weighing bottle (35 mm × 90 mm). The prepared film sample was covered in the mouth of the weighing bottle, sealed, and placed in a desiccator with distilled water at the bottom. The mass of the weighing bottle was measured every 24 h for 8 consecutive days. Each sample was measured three times and the results were averaged. The WVP of the film is calculated according to Equation (3):(3)$$\mathrm{WVP}=\frac{W}{t\times A}\times \frac{X}{\Delta P}$$

In the formula: WVP, water vapor permeability (g· mm/m^2^·d· kPa); W, total mass of the bottle after film sealing, (g); t, the testing time (s); A, the permeable film area (m^2^); X, the film thickness (m); and ΔP, Vapor pressure difference between two sides of the film (1583 Pa at 25 °C).

#### 2.4.3. Analysis of Film Color and Transparency

The L* value (brightness), a* value (red/green), and b* value (yellow/blue) of the film samples were determined by chromatic aberration meter (UPG-722 Visible spectrophotometer, Beijing Uber General Technology Co., Ltd., China). Before measurement, a standard plate calibration (L* = 103.98, a* = −5.80, and b* = 9.25) was used. Each sample was measured three times and the results were averaged.

The film samples were sheared into a rectangle (10 mm × 45 mm). The rectangular film was attached to the inner wall of the empty cuvette and the empty cuvette was used as the contrast. The absorbance value of the sample was measured at 600 nm. Each sample was measured three times and the results were averaged. The opacity was calculated according to Equation (4):(4)$$\mathrm{Opacity}=\frac{A_{600}}{X}$$

In the formula: A_600_ is the absorbance at 600 nm, and X is the film thickness (mm).

#### 2.4.4. Scanning Electron Microscopy (SEM)

A scanning electron microscope (S-4300, Hitachi, Japan) was used to observe the microstructure of the composite films surface and cross-section. The composite film was cut into small rectangles of 4 cm × 6 cm, broken up with liquid nitrogen, and then gold-plated by sputtering. The scanning voltage was 2.00 kV and the current was 64.0 µA. The surface and cross-section structures of the composite films were photographed and observed.

#### 2.4.5. Fourier Transform Infrared (FT-IR) Spectroscopy

To analyze the effect of GSE on the chemical structure of composite film, FT-IR spectra were obtained using FT-IR/NIR spectroscopy (Spotlight 400, Perkin Elmer Co., Waltham, MA, USA) according to the method of Wang et al. [25]. The composite film was placed on the ATR accessory, and the operation parameters of SEM were as follows: the test temperature was 25 °C, the wave number was 4000–650 cm^−1^, the resolution was 4 cm^−1^, and the scans were carried out 32 times.

#### 2.4.6. X-ray Diffraction (XRD)

The crystalline characteristics of films were analyzed via XRD (SmartLab, Rigaku Co., Japan), according to the method described by Guo et al. [19] with some modifications. The XRD (SmartLab, Rigaku Co., Tokyo, Japan) analysis of the films was carried out using the method of Ilyas et al. [16] using CuKα radiation, the XRD scan ranged from 5–80°(2θ), and the scan rate was 2°/min.

#### 2.4.7. Differential Scanning Calorimetry (DSC) Analysis

The thermal transition of components was measured using differential scanning calorimetry. About the 4 mg of the dried film samples were sealed in an aluminum crucible. The empty aluminum crucible acted as a blank control. Nitrogen was used as the protective gas. The protective gas flow rate was 20 mL/min. The heat-up rate was 10 °C/min. The differential scanning calorimetry analysis was performed in a temperature range of 20 °C to 250 °C.

### 2.5. Assay of the Antioxidant and Antibacterial Activity of Films

#### 2.5.1. Total Phenol Content

The total phenol content of the film was determined by the Folin–Ciocalteu method with slight modifications [26]. The film immersion solution was obtained by immersing 125 mg of the film sample in 15 mL of distilled water for 24 h. A 0.1 ml film immersion solution, 7 mL distilled water, and 0.5 mL folinol were added into a 50 mL conical flask in turn and shaken gently. After standing for 8 min, 1.5 mL of 10 wt% Na_2_CO_3_ solution and 0.9 mL of distilled water were added in turn. The composite solution was placed in a dark room to avoid light for 2 h. The absorbance of the mixture was measured by a UV spectrophotometer at 765 nm. A series of aqueous gallic acid solutions were prepared with concentrations ranging from 0 to 15 μg/mL. The absorbance at 765nm was measured as described above. The standard curve was drawn with the concentration of aqueous gallic acid as abscissa and its absorbance at 765 nm as ordinate. The equation y = 0.117x + 0.0171 (R^2^ = 0.9995) was obtained. Each sample was measured in triplicate and averaged. The total phenolic content of the samples was expressed as milligram gallic acid (GAE) equivalents per gram of dry matter (DW), GAEmg/DWg.

#### 2.5.2. DPPH Free Radical Scavenging Activity of Films

Ten milligrams of the film sample were mixed with DPPH methanol solution (1.5 mL, 0.2 mM). The mixture was placed into a dark chamber for the reaction. Light was avoided for 0.5 h at room temperature. After that, DPPH methanol solution was used as a blank control and the absorbance of the mixture was determined at 517 nm (UV). Each sample was measured three times and the results were averaged. To calculate the DPPH-free radical scavenging rate, the following is used:(5)$$k_{\mathrm{DPPH}}=\frac{A_{C}-A_{S}}{A_{C}}\times 100\%$$

In the formula: k_DPPH_ is the DPPH-free radical scavenging ability at 517 nm, A_c_ is the absorbance of the blank control at 517 nm, and A_s_ is the absorbance of the film samples at 517 nm.

#### 2.5.3. ABTS Free Radical Scavenging Activity of Films

The ABTS radical scavenging ability of the films was measured using the method of Riaz et al. [27] with a slight modification. A total of 10 g of the film sample was mixed with ABTS radical working solution and kept away from light for 6 min at room temperature. The absorbance of the mixture was measured at 734 nm with an ultraviolet spectrophotometer. An acetic acid buffer solution instead of film sample solution was used as a blank control. Each sample was measured three times and the results were averaged. The calculation formula of the ABTS-free radical scavenging rate is as follows:(6)$$k_{\mathrm{ABTS}}=\frac{A_{C}-A_{S}}{A_{C}}\times 100\%$$

In the formula: k_ABTS_ is the ABTS free radical scavenging ability at 734 nm, A_c_ is the absorbance of the blank control at 734 nm, and A_S_ is the absorbance of the film samples at 734 nm.

#### 2.5.4. Antimicrobial Activity

The antimicrobial activity of the composite films is determined by disk diffusion experiments. The expanded *E. coli* and *Staphylococcus aureus* were diluted 10-fold with sterile saline, respectively, as the initial bacterial solution. The composite films samples were cut into many circular sheets of 10mm, and the sheets were placed in a culture medium containing 0.1 mL of bacteria. The medium was inverted in a 37 ± 1 °C incubator and fostered for 24 h. The diameter of the inhibition zones was measured by a sliding caliper, and the size of the inhibition zone reflected the bacteriostatic effect of the composite films.

#### 2.5.5. Determination of Peroxide Value

An amount of 5 g solid lard was wrapped in the film, and the film was heat-sealed and placed in a blast-drying oven at 60 °C to accelerate oxidation. The samples were sampled at every 24 h interval and measured continuously for 7 d. The POV value was determined according to the method of Gao et al. [22]. Each sample was measured three times, and the results were averaged.

### 2.6. Statistical Analysis

All experiments were performed in triplicate independently and the results are expressed as the mean ± standard deviation (SD). Duncan’s test in the software SPSS 25 system was used for significance analysis. Origin 8.0 software was used for drawing. If *p* < 0.05, the difference was considered statistically significant.

## 3. Results and Discussion

### 3.1. Physical Properties of the CS/κC-GSE Films

#### 3.1.1. Mechanical Properties

The mechanical properties of composite films are expressed by tensile strength (TS) and elongation at the break (EB). The effects of the different concentrations of GSE on the mechanical properties of the film are shown in Table 1. The CS/κC film had the highest TS, reaching 9.07 MPa. Conversely, the CS/κC film had the lowest EB of only 22.37%. The EB of GSE composite membrane is higher than that of CS/κC membrane, and EB increases with the increase of GSE concentration. On the contrary, TS decreased with the increase of GSE concentration. The addition of GSE increased the intermolecular forces controlling the film [3]. The interaction of GSE and CA loosens the spatial structure of starch and thus gives CS/κC-GSE membrane good mechanical properties [27]. The decrease of TS may be because the interaction between starch and polymer molecules was weakened by the addition of GSE [28]. The increase of EB and decrease of TS suggested an enhancing degree of the plasticization of the polymers [29]. Similar studies also confirmed that the addition of plant extracts could improve the mechanical properties of starch-based films [30,31,32,33]. Bof et al. [34] developed starch–chitosan-based films incorporated with lemon essential oil (LEO) and grapefruit seed extract (GSE). They found that the addition of GSE increased the EB value of the films because of the interaction between GSE and the polymer matrix. It has been reported that the addition of GSE and ZEO essential oil (ZEO) to the chitosan film can improve the mechanical properties of the chitosan film. However, with the increase of ZEO concentration in the chitosan film, the EB decreased significantly, which may be due to the increase of the pore size of the film to create a possible breaking point [35]. Similarly, Söğüt et al. [28] recently found that the addition of GSE to gelatin films resulted in a significant reduction in TS. They attributed the phenomenon to polyphenolic compounds that impair protein interactions.

#### 3.1.2. Water Vapor Permeability (WVP)

Food packaging film should have the function of preventing moisture and packaged food from transferring to each other [36]. WVP is one of the best determinants of delaying water movement between the film and the surrounding environment to prolong the shelf life of food [20]. The WVP was an essential measure of the quality of food packaging. WVP is affected by polymer properties [37]. The WVP of the CS/κC-GSE film is shown in Table 1. The WVP of the CS/κC film was 1.08 g· mm/m^2^·d· kPa. The WVP of CS/κC film mixed with GSE is higher than that of the CS/κC film and increases with the increase of GSE concentration. Similar studies also confirmed that the addition of plant extracts could increase the WVP value of starch-based films [30,38]. They found that the presence of GSE may produce less crystalline film, leading to an increase in WVP. Söğüt et al. [39] found that the films containing GSE had high WVP values, indicating that plasticization may have occurred. Promsorn [40] also indicated increased WVP due to a higher degree of plasticization in starch-based films, which increased free volume and molecular mobility. Similar results were found by Wu et al. [41], who added GSE to TEMPO-oxidized nanocellulose (TNC) films and found an increase in the WVP of the films. They found that the film absorbs water from the air because of the loss of film structure caused by the addition of GSE. Basch et al. [30] found that the presence of Nisin (Nis) and Nis/potassium sorbate (KS) resulted in a significant increase in this parameter in all studied films, which they suggested could be responsible for the biological destructor matrix produced by antibiotics. In contrast, experiments by Tavares et al. [42] showed that the addition of carboxymethyl cellulose (CMC) polymer to cornstarch film decreased the WVP value because the formation of CMC and starch blends improved the water resistance of CMC to some extent.

#### 3.1.3. Color, opacity, and light transmittance of the film

As a food packaging film, the appearance and optical performance are critical indicators for measuring the quality of the film [43]. As shown in Table 2, the GSE was mixed in the CS/κC film to significantly change the color of the film. The color of CS/κC film was colorless and transparent. When GSE was added to the CS/κC film, a*, b*, and opacity increased significantly, which meant that the CS/κC film gradually turned yellow. In contrast, the L* of the film slowly decreased. The above results show that the amount of GSE added has a significant influence on the color and light transmittance of the film (*p* < 0.05). This may be due to the color of the GSE itself, leading to a significant change in the color of the film. Similarly, Söğüt et al. [28] found that GSE incorporation into chitosan (CH) films reduced the brightness of CH films due to phenolic interactions. Bof et al. [34] found that the darkening of GSE films was due to the effect of promoting light scattering. Moradi et al. [35] reported a decrease in the brightness of the films with the addition of polyphora essential oil and GSE, which they attributed to the lack of interaction between the polymers.

### 3.2. Scanning Electron Microscopy (SEM)

SEM was used to analyze the film’s microstructure, whose surface smoothness and pores could be determined by SEM [44]. Figure 1 is the analysis of the CS/κC films and the CS/κC-GSE films by SEM. The microstructure of CS/κC film was continuous, smooth, and homogeneous without any pores, cracks, or irregular phenomenon. Since carrageenan and glycerol are mixed well in the starch film, CS/κC film has a smoother slope [25]. The GSE was mixed into CS/κC film, and some speckles and irregular shapes appeared on the surface of the film. When the addition amount of GSE was 3%, the cross-section of the composite film was stratified. When the amount of GSE was 5%, the stratification phenomenon was more obvious, which may be due to the agglomeration of proanthocyanidins in GSE. No significant surface aggregation of the film at a low concentration of GSE. The irregular shape progressively appears in the film, the surface is gradually rough, and a small number of insoluble particles can be observed. This may be because when the amount of GSE added is too much, the solubility in the system becomes worse, leading to the appearance of insoluble particles. Cross-section images also show rough and non-homogeneous structures when adding GSE. This non-homogeneity reflected immiscibility between of the film components [45,46]. Such results are similar to those found by Rubilar et al. [38]. They studied and found that it may be due to the hydrophilicity of the GSE. Meanwhile, they found that GSE was evenly mixed with carvacrol in the cross-section. Chen et al. [30] observed the presence of particles observed after the addition of ethanolic extract of thyme (TH) to potassium sorbate film (KS) is caused by the accumulation of particles during drying.

### 3.3. Fourier Transform Infrared (FTIR) Spectroscopy

FT-IR spectral analysis can research the interaction between the functional groups and intermolecular interactions in the films [47]. Figure 2 represents the FT-IR spectra of the film. The CS/κC film showed more obvious absorption peaks at 3304, 2926, 1645, 1150, and 1014 cm^−1^. Among them, there is a broad absorption peak at 3200–3400 cm^−1^, related to the stretching vibration of the O-H [41]. Meanwhile, the peak value of GSE film was higher than that of CS/κC film, indicating that the content of hydroxyl group also increased with the increase of GSE concentration [48]. There seems to be a slight shifting of the O-H stretching bands towards lower wavenumbers, which suggested hydrogen bonding between hydroxyl groups of polymer components and additives (GSE) [45]. A peak of 2926 cm^−1^ observed in the CS/κC film can be attributed to the asymmetric tensile vibration of the CH_2_ bond, and similar bands were also observed in the GSE films [49]. The absorption peak at 1645cm^−1^ occurs because the asymmetric and symmetric expansion vibrations occur at C=O. The bands at 1440 and 1060 cm^−1^ are characteristic of the C-H deformation of the aromatic ring [50]. The band vibration of C-H and CH_2_ deformation was observed between 1200 and 1500 cm^−1^ [10]. The absorption peak is gradually apparent at 1150–1014 cm^−1^, which may be caused by bending the vibrations out of the C-H plane of the aromatics [51]. The absorption peak of 928 cm^−1^ is generated by bending vibrations outside the plane of the alkene [52]. When GSE was mixed into the composite film, the vibration absorption peak of -OH in the composite film was slightly red shifted to the high wave segment, indicating that GSE affected the intermolecular force of starch. This is similar to that found by Wu et al. [41], suggesting that there was a hydrogen bonding interaction among TNC, GSE, and AgNPs. As can be seen from FTIR, GSE can form hydrogen bonds with relevant functional groups in the composite film, reducing free hydrogen and forming hydrophilic bonds. Therefore, the mechanical properties and water isolation properties can be improved to a certain extent.

### 3.4. X-ray Diffraction (XRD)

The properties and structure of the composite film are closely related to the crystallinity. Crystallinity is an important parameter of composite film properties, which is usually closely related to the stability of the material [53]. The film XRD images are shown in Figure 3. The broad peaks of the CS/κC films near 2θ = 20.00° without obvious absorption peaks indicate that the CS/κC films are non-crystallized [22]. Three similar reflection peaks appeared at 19.06°,19.70°, and 20.30°. In contrast, a more pronounced diffraction peak appeared at 2θ = 19.78° in the addition of 1% GSE films. The new diffraction peaks become sharp and the intensity of the CS/κC-GSE films increases with the increase of GSE content. This indicates that GSE can improve the crystallinity of CS/κC film. Further confirmed is the good binding of GSE to the CS/κC film, but no other diffraction peaks of GSE were found [42]. It is worth noting that all of the composite films show an amorphous pattern similar to the CS/κC film at 2θ ≥ 25.00°. The diffraction peak intensity of the CS/κC-GSE films was significantly higher than that of the CS/κC films. The crystalline properties of the CS/κC films are affected by the content and composition of polyphenols in the extracts [54]. Similarly, Gao et al. [22] found a similar effect with the addition of black soy bark ethanol extract to corn starch. They suggest that the crystalline properties of polyphenol-rich CS-based films are influenced by the content and composition of polyphenols in the extract.

### 3.5. Differential Scanning Calorimetry (DSC) Analysis

The differential scanning calorimetric analysis of CS/κC-GSE film was carried out. Glass transition temperature (Tg) is an effective indicator of compatibility between polymers [55]. Tg shows an endothermic shift. The peak is a phase transition of the first order. The Tg curve CS/κC film has a large and broad absorption peak at 132.32 °C (Figure 4). With the addition of GSE, the broad absorption peak gradually becomes the significant absorption peak. The higher the GSE concentration, the more obvious the peak value and the lower the corresponding temperature. This is due to the interaction of the GSE with the polymer, which reduces the Tg peak and enhances the flexibility of the molecule [56]. The Tg content of 5% GSE films was higher than the Tg content of CS/κC films. It shows that the addition of GSE enhances the wear resistance and low temperature resistance of the film and also enhances the flexibility of the molecular chain [57,58]. The Tg curves of all the films showed a peak value, indicating that the films had a good compatibility. Aslaner et al. [55] found that the addition of GSE had a strong interaction in the polymer matrix. Similar results were obtained by Niazi et al. [56]. They found that Tg was reduced in the prepared citric acid plasticized thermoplastic starch (TPS) films because the plasticizer broke the hydrogen bonds between the starch molecules. Similar results were observed in the study by Wang et al. [59]. The decrease in Tg may be due to the impairment of the starch chain interaction by GSE and the increase of starch chain flexibility.

### 3.6. Biological Activity of the CS/κC-GSE Films

The higher the content of TPC in the film, the stronger the anti-oxidation ability of the film, which can effectively delay food spoilage [60]. The content of TPC of the CS/κC-GSE films is shown in Figure 5. The TPC of the CS/κC-GSE films was significantly higher than that of the CS/κC film, and the higher the concentration of GSE, the higher the TPC content. Similar studies have also confirmed that the addition of plant extracts can increase the TPC value of starch-based films [56,61]. Similar results were also found by Maroufi et al. [61] who added green tea extract to fish gelatin-based films and found that the plant extract effectively increased the TPC content of the films.

Free radical scavenging ability is also an important indicator for measuring the quality of active packaging films [22]. Figure 6 shows that the CS/κC film of ABTS and DPPH free radical scavenging rate were the lowest. When GSE was added to the CS/κC film, the scavenging rates of the DPPH and ABTS radicals of the CS/κC film increased with the increase of GSE concentration. Similarly, some studies have found that adding natural extracts into the film can enhance the free radical scavenging rate of the film [24,27]. Roy et al. [24] found that the addition of Alizarin and grapefruit seed extract to chitosan films had similar results. They found that the main scavenging effect is the presence of phenyl groups in alizarin and the presence of anthocyanin substances in the GSE. Similar findings were also reported by Söğüt et al. [27]. They found that the GSE could inhibit the oxidation of chicken breasts.

POV refers to measuring the concentration of hydrogen peroxide formed during the initial phase of lipid oxidation [22]. The higher the POV value, the higher the oxygen transmittance. Figure 7 shows the change of POV over 7 days of film storage. The POV of each index was proportional to the heating time. The POV of coated lard was significantly lower than that of uncoated lard, indicating that the composite film had an inhibitory effect on lard oxidation. With the increase of GSE concentration, the inhibitory effect of GSE was enhanced. This may be due to the microstructure and compactness of the composite film added with GSE were affected. This blocks the passage of some oxygen and slows down the oxidation rate of lard [62]. However, the POV of 5% CS/κC-GSE film on the 7th day was higher than that of the CS/κC film. The main reason for this difference was that the surface roughness of the composite film was caused by GSE, which affected the compactability and barrier properties of the composite film [31]. Similar conclusions were reached by Rubilar et al. [38], they concluded that adding carvacrol and GSE to the film could effectively increase the barrier performance of the film to oxygen.

The antimicrobial properties of thin films are crucial for food packaging films. Today, to judge the antimicrobial activity of film, we usually evaluate the antibacterial activity of film samples by observing the size of the inhibitory region of food-derived bacteria [20,63]. In this experiment, *E. coli* and *S. aureus* were selected to determine the antibacterial activity of the film. Table 3 shows the analysis of the antibacterial properties of the CS/κC films and CS/κC-GSE films. The CS/κC film had a small inhibitory effect on *S. aureus* and *E. coli* because of the effect of CA in the CS/κC film [64]. The antibacterial effect of GSE added to CS/κC film on *E. coli* and *S. aureus* gradually increased, and the antibacterial effect was positively correlated with the concentration of GSE. CS/κC-GSE significantly inhibited foodborne bacterial growth during antimicrobial analysis because of the large number of polyphenols present in the GSE. Polyphenols can not only increase the permeability of cell film and cause the inhibition of energy metabolism but also interfere with the growth and reproduction of microorganisms. So as to inhibit the growth of microorganisms and improve the antibacterial performance of CS/κC composite film [65]. Similarly, Lim et al. [66] found that the packaging films of GSE can also better delay the growth of L. monocytogenes in soft cheese. Bof et al. [34] found that in the case of GSE, it was effective against Gram+ and Gram− only when it was applied directly to the paper disks.

## 4. Conclusions

The CS/κC-GSE composite film was prepared by mixing biodegradable materials such as GSE, κC and CS. Hydrogen bonds were formed between starch and GSE, which significantly increased the elongation at break and WVP of starch film. There was higher total phenolic content in CS/κC-GSE films compared to CS/κC films, whereas 5% CS/κC-GSE film was the most effective on DPPH and ABTS. POV experiments showed that the oxidation rate of oil was slowed down by the addition of GSE, and the antioxidant capacity of 5% CS/κC-GSE film was significantly increased. The antibacterial activity of CS/κC-GSE film against *E. coli* and *S. aureus* was higher than that of CS/κC film, and the antioxidant barrier of the film was significantly increased by 5% GSE film. In conclusion, the CS/κC-GSE film prepared in this study is non-toxic, biodegradable, and has good antioxidant and antimicrobial properties. Therefore, the CS/κC-GSE film is not only a safe and environmentally friendly active packaging material but can also be used as an active packaging film to extend the shelf life of foods. In this study, CS/κC-GSE film active packaging is only reflected in the good antioxidant properties. In the future, we will apply the film in different meat products to provide a theoretical basis for the preservation of meat products in the future.

## Figures and Tables

**Figure 1 polymers-14-04857-f001:**
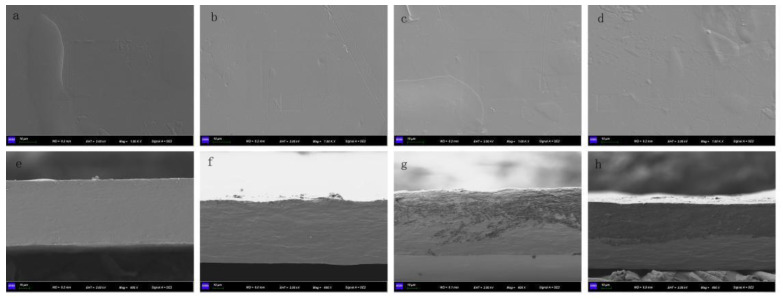
SEM micrographs of the surface and cross-section of films. (**a**) surface and (**e**) cross-section of CS/κC film; (**b**) surface and (**f**) cross-section of 1% CS/κC-GSE film; (**c**) surface and (**g**) cross-section of 3% CS/κC-GSE film; (**d**) surface and (**h**) cross-section of 5% CS/κC-GSE film.

**Figure 2 polymers-14-04857-f002:**
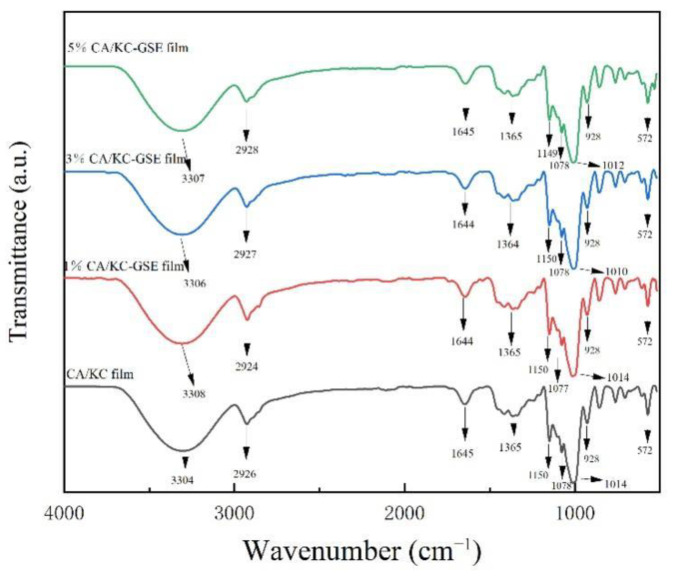
FT-IR spectra of CS/κC, 1% CS/κC-GSE, 3% CS/κC-GSE, and 5% CS/κC-GSE films.

**Figure 3 polymers-14-04857-f003:**
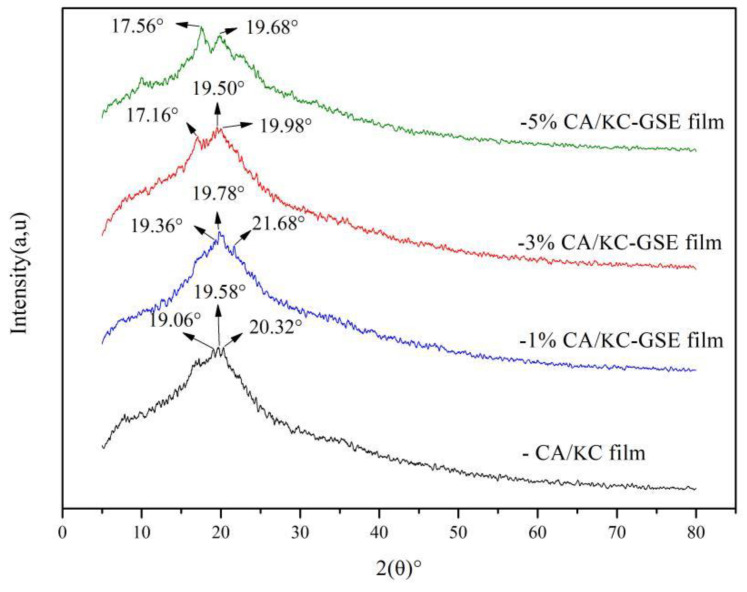
XRD patterns of CS/κC, 1% CS/κC-GSE, 3% CS/κC-GSE, and 5% CS/κC-GSE films.

**Figure 4 polymers-14-04857-f004:**
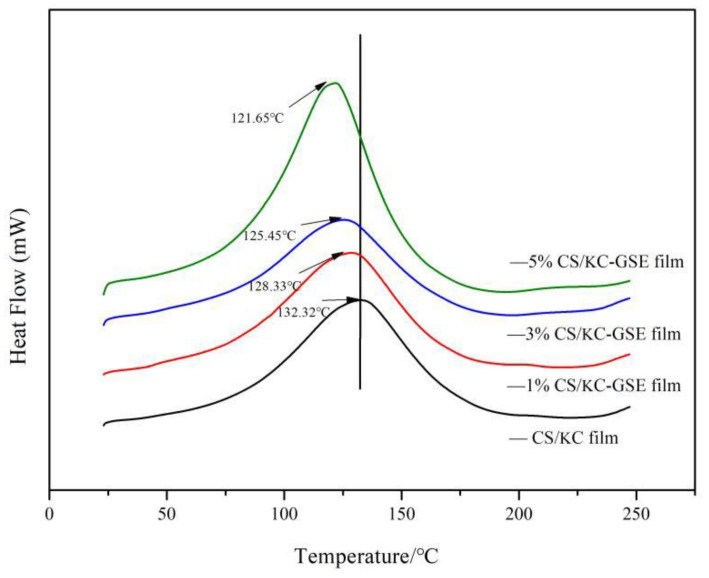
DSC curves of CS/κC, 1% CS/κC-GSE, 3% CS/κC-GSE, and 5% CS/κC-GSE films.

**Figure 5 polymers-14-04857-f005:**
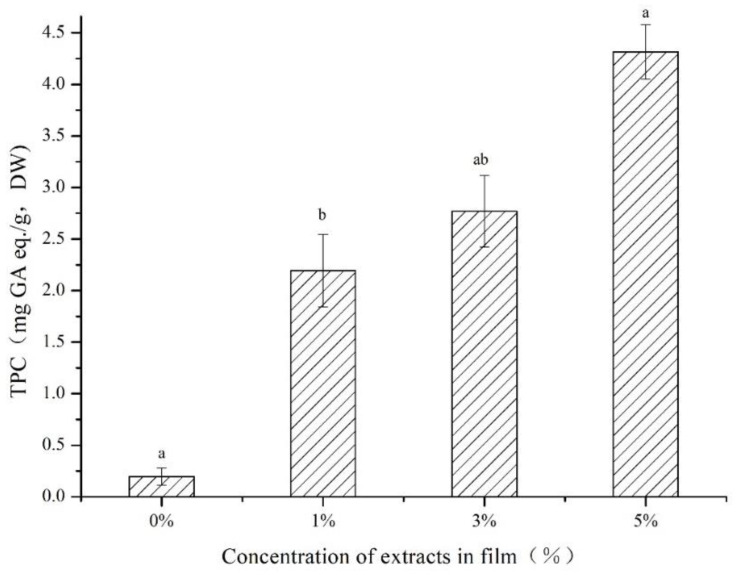
Total phenolic content of the CS/κC, 1% CS/κC-GSE, 3% CS/κC-GSE, and 5% CS/κC-GSE films. a, b Values are given as mean ± standard deviation. Different letters in the same line indicate significantly different (*p* < 0.05).

**Figure 6 polymers-14-04857-f006:**
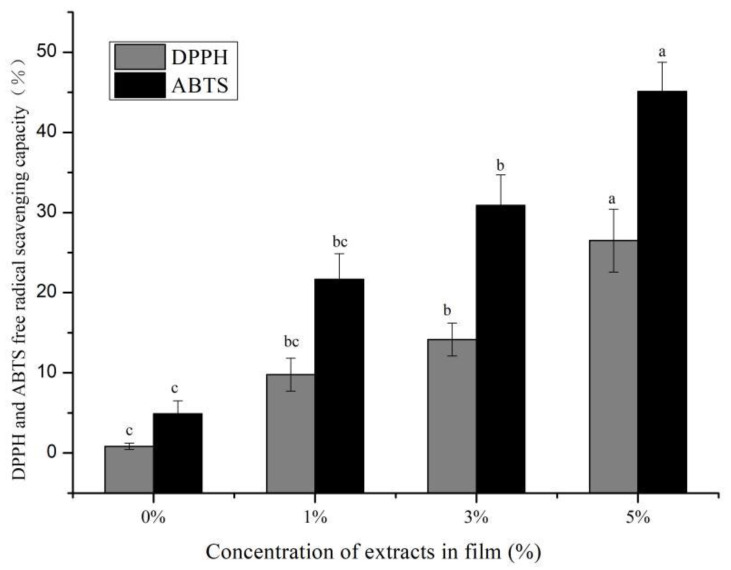
Scavenging activities of the CS/κC complex films on DPPH and ABTS radicals. a–c Values are given as mean ± standard deviation. Different letters in the same line indicate significantly different (*p* < 0.05).

**Figure 7 polymers-14-04857-f007:**
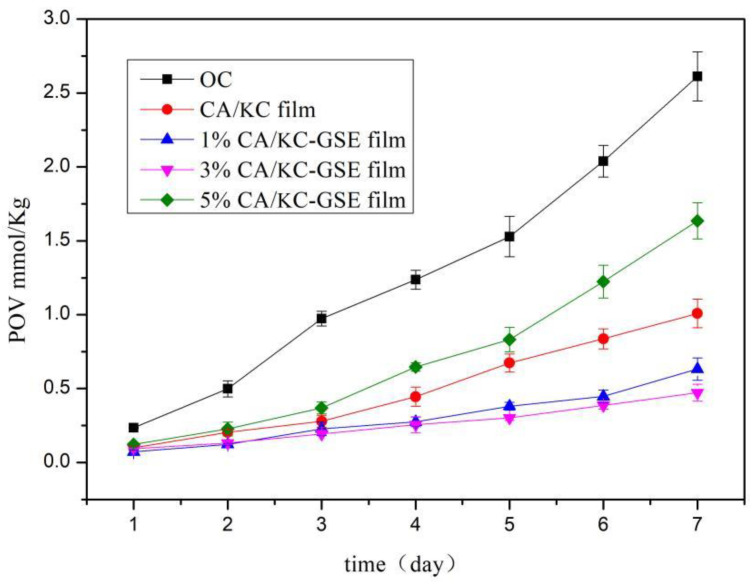
POV of lard packed by CS/κC-based films with a different concentration of GSE.

**Table 1 polymers-14-04857-t001:** Physical properties of CS/κC-GSE films with a different concentration of GSE.

Films Sample	Tensile Strength (MPa)	Elongation at Break (%)	WVP (g·mm/m^2^·d kPa)
CS/κC film	9.07 ± 0.45 ^a^	22.37 ± 0.98 ^c^	1.08 ± 0.04 ^c^
1% CS/κC-GSE film	5.34 ± 0.31 ^b^	25.94 ± 0.97 ^bc^	1.25 ± 0.02 ^bc^
3% CS/κC-GSE film	4.27 ± 0.41 ^bc^	29.92 ± 1.94 ^b^	1.38 ± 0.04 ^b^
5% CS/κC-GSE film	3.50 ± 0.27 ^c^	36.87 ± 2.08 ^a^	1.58 ± 0.03 ^a^

^a–c^ Values are given as mean ± standard deviation. Different letters in the same line indicate significantly different (*p* < 0.05).

**Table 2 polymers-14-04857-t002:** Color, optical properties, and opacity of CS/κC-GSE films with different concentrations of GSE.

Film Sample	L*	a*	b*	Opaqueness s/%	Picture
CS/κC film	90.24 ± 0.33 ^a^	−1.22 ± 0.04 ^d^	−0.18 ± 0.08 ^d^	1.06 ± 0.13 ^d^	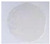
1% CS/κC-GSE film	85.74 ± 0.50 ^b^	1.70 ± 0.20 ^c^	3.64 ± 0.22 ^c^	2.25 ± 0.43 ^c^	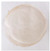
3% CS/κC-GSE film	81.22 ± 0.73 ^c^	4.18 ± 0.32 ^b^	6.96 ± 0.35 ^b^	3.03 ± 0.14 ^b^	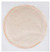
5% CS/κC-GSE film	77.08 ± 1.05 ^d^	6.74 ± 0.48 ^a^	10.90 ± 0.64 ^a^	3.76 ± 0.18 ^a^	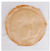

^a–d^ Values are given as mean ± standard deviation. Different letters in the same line indicate significantly different (*p* < 0.05).

**Table 3 polymers-14-04857-t003:** Antibacterial activity of CS/κC and CS/κC-GSE films against *E. coli* and *S. aureus*.

Extract Concentration	Diameter of the Bacteriostatic Circle (mm)	
	*Escherichia coli*	*Staphylococcus aureus*
CS/κC film	7.06 ± 0.03 ^d^	7.02 ± 0.05 ^d^
1% CS/κC-GSE film	9.62 ± 0.12 ^c^	10.04 ± 0.10 ^c^
3% CS/κC-GSE film	10.96 ± 0.28 ^b^	11.02 ± 0.22 ^b^
5% CS/κC-GSE film	12.78 ± 0.42 ^a^	13.52 ± 0.66 ^a^

^a–d^ Values are given as mean ± standard deviation. Different lowercase letters in the same column indicate significantly different (*p* < 0.05).

## Data Availability

The data presented in this study are available on request from the corresponding author.

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
