# Peer review of "Biodegradable Active Packaging Material Containing Grape Seed Ethanol Extract and Corn Starch/κ-Carrageenan Composite Film"

_polymers, 2022, doi:10.3390/polym14224857_

Round 1

Reviewer 1 Report

 The authors studied preparation and characterization of starch/ carrageenan/grape seed extract films for applications in packaging.

The topic has both academic and industrial interest. However, the following comments should be answered/responded to before considering publication.

In Results and Discussion

-          Please add Young's modulus for tensile properties table (Table 1) with suitable discussion.

-          Please revise the unit of WVP in the same table (x 1010)

-          It is noted in Table three that the blank film (without grape seed extract) has antimicrobial activity too. The author needs to comment on that with good references; what are the components in the blank film that can make it antibacterial?

Reviewer 2 Report

Overall, the study is compact and well-written. It may be recommended after revision.

1. Abstract should start with the objective definition.

2. The last paragraph of the introduction needs improvement. The novelty should be properly mentioned.

3. Line 283, "GSE and к -carrageenan enhance the intermolecular interaction", explain in detail.

4. Kindly check the manuscript and reword the long sentences as they are difficult to read and grasp.

5.  Kindly include the future scope of the study in the conclusion.

Reviewer 3 Report

Wang et al investigated "biodegradable active packaging material containing grape seed ethanol extract and corn starch/к-carrageenan composite film.

The manuscript is written well and the experiments were conducted well however the subsequent points need to be revised and please reply the answer to me again:

The abstract needs to rewritten again to include numbers of results instead of theoretical explanation.

The introduction needs to add some updated information literature in the same topic like

-          Production of Bio‑composite Films from Gum Arabic and Galangal

Extract to Prolong the Shelf Life of Agaricus bisporus

-          Characterisation, rheological properties and immunomodulatory

efficiency of corn silk polysaccharides.

-          Production and Characterization of Sodium Alginate/Gum Arabic

Based Films Enriched with Syzygium cumini Seeds Extracts for Food

Application.

If it is available to add images for petri plates that prove the inhibition of E.Coli and Staphylococcus aureus.

The section 2.5.5 is written very poorly. Please, rewrite it.

In section 3.1.1, please add the increasing or decreasing percentages numbers.

In section 3.1.2, please at the add explain the importance of WVP in food wrapping.

In Table 2. The L* values are decreased. It is supposed to increase because the addition of extracts increased the opaqueness. I feel that it is wrong calculation. And else, a* values should increase.

The SEM images are not clear. Please, add the same magnification power of all images.

Round 2

Reviewer 1 Report

The reviewer asked for some revision but the response to the comments could not be found in the submitted manuscript although the authors mentioned in their response letter that it is done as follows:

 The reviewer comment: Please add Young's modulus for tensile properties table (Table 1) with suitable discussion.   

Response of the authors: It has been added.

 But there is nothing is found in the submitted manuscript regarding the Young's modulus

 The reviewer comment: Please revise the unit of WVP in the same table (x 1010)

 Response of the authors: It has been revised. It is gm-1s-1Pa

 But the unit in the experimental part is different from that in the results in Table 1.

 Reviewer comment:  It is noted in Table three that the blank film (without grape seed extract) has antimicrobial activity too. The author needs to comment on that with good references; what are the components in the blank film that can make it antibacterial?

 Response of the authors: It has been added.

 The authors didn't answer the question of the authors. In addition, the sentence that the authors added is not clear. (The composite film has inhibitory effect on S. aureus and E. coli, but the inhibitory effect is not obvious. This is  because the carrageenan in the composite membrane played an inhibitory role on both  bacteria [64]).

Which film the authors mean?
